# Replicability of sight word training and phonics training in poor readers: a randomised controlled trial

G McArthur, S Kohnen, K Jones, P Eve, E Banales, L Larsen and A Castles

Department of Cognitive Science, ARC Centre of Excellence in Cognition and its Disorders, Macquarie University, NSW, Australia

## ABSTRACT

Given the importance of effective treatments for children with reading impairment, paired with growing concern about the lack of scientific replication in psychological science, the aim of this study was to replicate a quasi-randomised trial of sight word and phonics training using a randomised controlled trial (RCT) design. One group of poor readers ($N = 41$) did 8 weeks of phonics training (i.e., phonological decoding) and then 8 weeks of sight word training (i.e., whole-word recognition). A second group did the reverse order of training. Sight word and phonics training each had a large and significant valid treatment effect on trained irregular words and word reading fluency. In addition, combined sight word and phonics training had a moderate and significant valid treatment effect on nonword reading accuracy and fluency. These findings demonstrate the reliability of both phonics and sight word training in treating poor readers in an era where the importance of scientific reliability is under close scrutiny.

Corresponding author
G McArthur,
genevieve.mcarthur@mq.edu.au

## INTRODUCTION

Around 5% of children have a significant reading impairment despite normal reading instruction, normal intelligence, and the absence of any known neurological or psychological problems. This condition—often called developmental dyslexia (*Hulme & Snowling, 2009*)—not only affects a child's academic achievements, but increases their risk for anxiety, depression, conduct disorder, and hyperactivity (*Carroll et al., 2005*). Thus, it is critical to discover how to treat poor readers as early and effectively as possible.

To date, most treatment trials done with poor readers have looked at the effects of "phonics" reading programs, which teach children to learn to explicitly "phonologically decode" words by converting graphemes (i.e., letters or letter clusters; e.g., SH, I, P) into sounds (e.g., sh as in *cash*, i as in *in*, and p as in *pin*) and then blend those sounds into a word (*ship*). Since the turn of the century, at least three systematic reviews and meta-analyses have examined the effect of phonics training in poor readers. In 2001, *Ehri et al. (2001)* reported that phonics training, administered either alone or in combination with other types of training (e.g., phoneme awareness), had a moderate effect on poor

readers' explicit phonological decoding, but a small effect on their word reading. In 2012, *McArthur et al. (2012)* reported that "specific" phonics training, which focused on explicit phonological decoding with minimal training in other skills, had a large effect on poor readers' explicit phonological decoding, a moderate effect on their word reading, and a small-to-moderate effect on their grapheme-phoneme correspondence (GPC) knowledge. In 2014, *Galuschka et al. (2014)* reported that specific phonics training had a small but significant effect on reading measures averaged across different reading tests. Considered together, the outcomes of these reviews suggest that phonics training in poor readers might have significant and large effects on reading measures that depend heavily on explicit phonological decoding, but weaker effects on reading measures that depend on other skills, such as "sight word reading" (i.e., recognizing whole words from orthographic memory) and "reading comprehension" (i.e., understanding the meaning of written texts).

The outcomes of these systematic reviews align with the two widely used cognitive models of word reading: the dual route model and the triangle model (e.g., *Coltheart et al., 2001*; *Plaut et al., 1996*). According to both models, printed letters trigger cognitive processes relating to letter identification, the outputs of which are fed through to two pathways: (1) a "sublexical route" (dual route model)/ "phonological pathway" (triangle model); and (2) a "lexical route" (dual route model)/"semantic pathway" (triangle model). The sublexical/phonological pathway, which includes links between orthography and phonology, makes a greater contribution to reading "regular" words (i.e., real words that can be read correctly via explicit phonological decoding, e.g., *ship*) and "nonwords" (i.e., nonsense words that can be read accurately via explicit phonological decoding, e.g., *shap*). In contrast, the lexical/semantic pathway, which has links between orthography, phonology, and semantics, makes a greater contribution to reading "irregular" words (i.e., words that cannot be read accurately via explicit phonological decoding alone, e.g., *yacht*).

Both the dual route and triangle models predict that phonics training should have its largest impact on the sublexical/phonological pathway, and hence the ability to read regular words and nonwords. This prediction is supported by the aforementioned systematic reviews by *Ehri et al. (2001)*, *McArthur et al. (2012)* and *Galuschka et al. (2014)* that suggest phonics training in poor readers has its largest effect on reading measures that depend most heavily on explicit phonological decoding (e.g., reading regular words and nonwords).

The dual route and triangle models further predict that training sight word reading (i.e., recognizing whole words from orthographic memory) should have its largest effect on the lexical/semantic pathway, and hence the ability to read irregular words. Unfortunately, there is little empirical data to test this prediction since most studies that examined the effects of sight word training on reading have included both regular and irregular words as training stimuli. The inclusion of regular words is problematic because, as hypothesized by the dual route and triangle models, regular words can be read with the sublexical/lexical pathway. Thus, the inclusion of regular words in sight word training obscures the true effect of training the lexical/semantic pathway. In order to test this effect, it is important to

employ "specific sight word training" that focuses training on recognizing written irregular words "by sight" (i.e., from orthographic memory).

To our knowledge, only one controlled trial has investigated the effect of specific sight word training in poor readers. *McArthur et al. (2013a)* gave three groups of children with poor reading different orders of sight word and phonics training. Group 1 ($N = 36$) received 8 weeks of "specific phonics training" (i.e., training reading via grapheme-phoneme correspondence (GPC) rules alone) followed by 8 weeks of specific sight word training (training the recognition of irregular words from orthographic memory). Group 2 ($N = 26$) received the same training in reverse order. Group 3 received "mixed" training that comprised the phonics training and sight word training on alternate days for two 8-week periods ($N = 32$). The outcomes revealed that: (1) specific sight word training had large and significant valid treatment effects on trained irregular words, untrained irregular words, word reading fluency, and word reading comprehension, as well as a moderate and significant treatment effect on nonword reading accuracy, and no treatment effect on nonword reading fluency; (2) specific phonics training had large and significant "valid" treatment effects (i.e., significantly larger than test-retest effects) on trained and untrained irregular words, word and nonword reading fluency, and reading comprehension, as well as a moderate-to-large effect on nonword reading accuracy; and (3) order of training (i.e., phonics-then-sight words; sight words-then-phonics; mixed) had an effect on untrained irregular word reading (significantly better after phonics-then-sight word training, than the reverse) but not on trained irregular words, nonword reading accuracy or fluency, word reading fluency, or reading comprehension (see Table 1 for a summary of the effects found by McArthur et al. compared to the current study).

Finding (1) was exciting because it showed, for the first time in a controlled group trial (albeit quasi-randomised), that specific sight word training has significant and large treatment effects in children with poor reading. Finding (2) was reassuring since it supported conclusions of the aforementioned meta-analyses that reported moderate to large phonics effects on some reading skills in poor readers. Finding (3) was puzzling since it appears to be often assumed by clinicians and researchers (despite the absence of empirical evidence) that poor readers should be taught explicit phonological decoding prior to sight word reading (*Chall, 1967*).

Given the importance of finding reliably effective treatments for poor readers, paired with growing concern about lack of replication in psychological and cognitive scientific research (e.g., *Drotar, 2010*; *Ioannidis, 2012*; *Pashler & Harris, 2012*; *Wagenmakers et al., 2012*), the aim of the current study was to test the replicability of McArthur et al.'s quasi-randomised controlled reading treatment trial. To this end, we used randomised controlled trial (RCT) that closely replicated the methods of McArthur et al. to test the replicability of key findings (1)–(3) outlined above.

## MATERIALS AND METHODS

The Macquarie University Human Research Ethics Committee (Ref: 5201200852) approved the methods outlined below. All children and their parents gave their informed

**Peer**J

Table 1 Training effects in *McArthur et al. (2013a)* and the current study for Group 1 and Group 2. T1T2, T1T3 and T1T4 represent gains in raw scores from Test 1 (before training) to Test 2 (after 8 weeks of no training), Test 3 (after the first 8 weeks of phonics in Group 1 or sight word training in Group 2), and Test 4 (after 16 weeks of training), respectively. Effect sizes (ES; Cohen's *d*) in bold indicate training gains significantly larger than T1T2. ESs of 0.3, 0.5, and 0.8 were considered small (S), medium (M), and large (L), respectively.

| | Group 1 | | | | | | Group 2 | | | | | |
| | McArthur et al. ($N = 36$) | | | Current ($N = 41$) | | | McArthur et al. ($N = 36$) | | | Current ($N = 44$) | | |
| | M | SD | ES | M | SD | ES | M | SD | ES | M | SD | ES |
|---|---|---|---|---|---|---|---|---|---|---|---|---|
| **Trained irregular word accuracy** | | | | | | | | | | | | |
| T1T2 | 1.06 | 1.72 | 0.6 (M) | 0.59 | 2.90 | 0.2 (S) | 1.31 | 2.54 | 0.5 (M) | 0.98 | 2.37 | 0.4 (S-M) |
| T1T3 | 2.67 | 2.12 | **1.3 (L)** | 2.08 | 2.98 | **0.7 (M-L)** | 5.25 | 3.52 | **1.5 (L)** | 2.73 | 2.61 | **1.0 (L)** |
| T1T4 | 5.14 | 3.32 | **1.6 (L)** | 3.39 | 3.25 | **1.0 (L)** | 5.14 | 4.26 | **1.2 (L)** | 3.39 | 2.44 | **1.4 (L)** |
| **Untrained irregular word accuracy** | | | | | | | | | | | | |
| T1T2 | 1.08 | 1.57 | 0.7 (M-L) | 1.34 | 2.55 | 0.5 (M) | 1.31 | 1.80 | 0.7 (M-L) | 1.39 | 2.01 | 0.7 (M-L) |
| T1T3 | 2.08 | 1.76 | **1.2 (L)** | 2.23 | 2.81 | **0.8 (L)** | 2.53 | 2.02 | **1.2 (L)** | 2.11 | 2.42 | **0.9 (L)** |
| T1T4 | 3.69 | 2.51 | **1.5 (L)** | 2.68 | 2.35 | **1.1 (L)** | 2.39 | 2.35 | **1.0 (L)** | 2.86 | 2.61 | **1.1 (L)** |
| **Nonword accuracy** | | | | | | | | | | | | |
| T1T2 | 1.28 | 3.70 | 0.4 (S-M) | 0.07 | 4.33 | 0.0 (S) | 0.17 | 3.79 | 0.0 (S) | −0.88 | 4.26 | −0.2 (S) |
| T1T3 | 2.75 | 4.12 | **0.7 (M-L)** | 1.78 | 6.21 | 0.3 (S) | 1.42 | 3.96 | **0.4 (S-M)** | −0.37 | 3.55 | −0.1 (S) |
| T1T4 | 3.00 | 3.96 | **0.8 (L)** | 2.17 | 5.14 | **0.4 (S-M)** | 3.64 | 4.89 | **0.7 (M-L)** | 1.23 | 3.26 | **0.4 (S-M)** |
| **Nonword fluency** | | | | | | | | | | | | |
| T1T2 | 2.03 | 4.55 | 0.4 (S-M) | −0.66 | 3.98 | −0.2 (S) | 1.78 | 4.70 | 0.4 (S-M) | 0.48 | 4.32 | 0.1 (S) |
| T1T3 | 3.72 | 4.69 | **0.8 (L)** | 1.17 | 4.85 | 0.2 (S) | 3.08 | 4.11 | 0.8 (L) | 1.00 | 3.94 | 0.2 (S) |
| T1T4 | 4.17 | 4.78 | **0.9 (L)** | 1.44 | 4.66 | 0.3 (S) | 3.03 | 5.05 | 0.6 (M) | 2.86 | 4.28 | **0.7 (M-L)** |
| **Sight word fluency** | | | | | | | | | | | | |
| T1T2 | 3.97 | 5.41 | 0.7 (M-L) | 2.66 | 6.28 | 0.4 (S-M) | 3.25 | 7.38 | 0.4 (S-M) | 2.64 | 4.83 | 0.6 (M) |
| T1T3 | 6.69 | 5.70 | **1.2 (L)** | 5.22 | 5.85 | **0.9 (L)** | 4.42 | 5.03 | **0.9 (L)** | 6.61 | 4.77 | **1.4 (L)** |
| T1T4 | 7.33 | 7.68 | **1.0 (L)** | 8.83 | 4.99 | **1.8 (L)** | 9.53 | 11.26 | **0.8 (L)** | 6.73 | 5.81 | **1.2 (L)** |
| **Reading comprehension** | | | | | | | | | | | | |
| T1T2 | 1.83 | 2.96 | 0.6 (M) | 1.17 | 2.19 | 0.5 (M) | 1.89 | 3.00 | 0.6 (M) | 1.23 | 2.53 | 0.4 (S-M) |
| T1T3 | 3.53 | 3.13 | **1.1 (L)** | 1.98 | 2.43 | **0.8 (L)** | 3.56 | 3.97 | **0.9 (L)** | 1.39 | 2.13 | 0.6 (M) |
| T1T4 | 4.78 | 4.32 | **1.1 (L)** | 2.51 | 2.48 | **1.0 (L)** | 4.22 | 4.28 | **1.0 (L)** | 1.98 | 2.62 | **0.8 (L)** |

consent to participate in this RCT. Children were continuously recruited into the study between January 2011 and June 2013 (i.e., children were not all tested and trained at the same point in time). Since it took 6 months for a child to complete the study, the last child completed the last test session (Test 4) in December 2013. This trial is registered with the Australian New Zealand Clinical Trials Registry (ANZCTR; 12608000454370). Methodological differences (all minor) between *McArthur et al. (2013a)* and the current study are outlined in parentheses.

## Trial design

All children completed screening and outcome measures at Test 1 (see Fig. 1). After 8 weeks of no training, they returned to do the outcome measures (Test 2) to index "non-treatment gains" (i.e., due to test-retest effects, test situation familiarity, regression to the mean,

|  | Group 1<br>N = 41 | Group 2<br>N = 44 |
|---|---|---|
| **Test 1**<br>2-3 hours | Screening measures<br>Outcome measures | Screening measures<br>Outcome measures |
| **No training**<br>8 weeks | No training | No training |
| **Test 2**<br>2-3 hours | Outcome measures | Outcome measures |
| **Train 1**<br>8 weeks | **Phonics** | **Sight words** |
| **Test 3**<br>2-3 hours | Outcome measures | Outcome measures |
| **Train 2**<br>8 weeks | **Sight words** | **Phonics** |
| **Test 4**<br>2-3 hours | Outcome measures | Outcome measures |

**Figure 1  Testing and training phases for each group.** The order of testing and training phases completed by the two groups.

maturation, and a "test-related Hawthorne effect" resulting from an awareness of being tested multiple times on similar outcome measures). Group 1 did 8 weeks of phonics training (and then Test 3) followed by 8 weeks of sight word training (and then Test 4). Group 2 did the same training in the reverse order. In the analysis, we controlled for non-treatment gains when comparing phonics training to sight word training, and when comparing phonics-then-sight word training versus sight word-then-phonics training. (Note: unlike *McArthur et al. (2013a)*, we did not include a "mixed" group since this group showed no advantage over groups 1 and 2 in McArthur et al.)

## Participants

In line with *McArthur et al. (2013a)*, this study recruited a typical "mixed" sample of poor readers from the community. Children were aged from 7 to 12; scored below the average range for their age (i.e., had a z score lower than −1.0, which represents the lowest 16% of readers) on the Castles and Coltheart 2 (CC2) irregular-word reading test and/or nonword reading test (*Castles et al., 2009*; see below); had no history of neurological or sensory impairment as indicated on a background questionnaire; and used English as their primary language at school and at home (see Screening Tests below). This resulted in a sample that was very similar in age, nonverbal IQ, and irregular word reading as McArthur et al. (see Table 2). However, the mean CC2 nonword reading scores for groups 1 and 2 in McArthur et al. (−1.50 and −1.27, respectively) were higher than in the current study (−1.66 and −1.62, respectively). Similarly, the mean CC2 regular word reading scores for groups 1 and 2 in McArthur et al. (−1.41 and −1.29, respectively) were higher than

**Table 2 Screening and outcome measures.** Means (M) and standard deviations (SD) for the screening and outcome measures.

| | | Group 1 | | Group 2 | |
|---|---|---|---|---|---|
| | | M | SD | M | SD |
| **Screening** | Age (years) | 9.53 | 1.51 | 9.58 | 1.45 |
| | Non-verbal IQ (s) | 97.02 | 15.75 | 97.57 | 16.45 |
| | CC2 Irregular words (z) | −1.42 | 0.65 | −1.37 | 0.75 |
| | CC2 Nonwords (z) | −1.66 | 0.57 | −1.62 | 0.67 |
| | CC2 Regular words (z) | −1.61 | 0.54 | −1.57 | 0.46 |
| **TT** | Sight word training (h) | 14.46 | 3.66 | 14.89 | 3.66 |
| | Phonics training (h) | 14.53 | 3.29 | 14.37 | 2.90 |
| **Test 1** | Trained irregular accuracy (r) | 12.59 | 7.28 | 13.64 | 8.06 |
| | Untrained irregular accuracy (r) | 11.34 | 7.31 | 12.59 | 8.44 |
| | Nonword reading accuracy (r) | 9.93 | 7.11 | 12.65 | 7.05 |
| | Nonword reading fluency (r) | 11.34 | 8.16 | 11.82 | 8.53 |
| | Word reading fluency (r) | 42.88 | 16.79 | 43.16 | 18.65 |
| | Reading comprehension (r) | 12.32 | 5.21 | 12.32 | 6.12 |
| **Test 2** | Trained irregular accuracy (r) | 13.17 | 7.87 | 14.61 | 8.23 |
| | Untrained irregular accuracy (r) | 12.68 | 7.80 | 13.98 | 8.77 |
| | Nonword reading accuracy (r) | 10.00 | 7.18 | 12.02 | 7.65 |
| | Nonword reading fluency (r) | 10.68 | 7.72 | 12.30 | 8.48 |
| | Word reading fluency (r) | 45.54 | 15.82 | 45.80 | 19.00 |
| | Reading comprehension (r) | 13.49 | 5.06 | 13.55 | 5.57 |
| **Test 3** | Trained irregular accuracy (r) | 14.90 | 7.38 | 16.36 | 8.34 |
| | Untrained irregular accuracy (r) | 13.82 | 7.82 | 14.70 | 8.48 |
| | Nonword reading accuracy (r) | 11.71 | 7.83 | 12.45 | 7.73 |
| | Nonword reading fluency (r) | 12.51 | 8.44 | 12.82 | 8.96 |
| | Word reading fluency (r) | 48.10 | 16.39 | 49.77 | 18.81 |
| | Reading comprehension (r) | 14.29 | 4.56 | 13.70 | 5.50 |
| **Test 4** | Trained irregular accuracy (r) | 15.98 | 7.02 | 17.02 | 8.02 |
| | Untrained irregular accuracy (r) | 14.02 | 7.63 | 15.45 | 8.67 |
| | Nonword reading accuracy (r) | 12.10 | 7.90 | 14.16 | 7.95 |
| | Nonword reading fluency (r) | 12.78 | 8.58 | 14.68 | 8.49 |
| | Word reading fluency (r) | 51.71 | 16.80 | 49.89 | 19.32 |
| | Reading comprehension (r) | 14.83 | 4.49 | 14.30 | 5.16 |

**Notes.**

CC2, Castles and Colthart reading tests (*Castles et al., 2009*); TT, time training; s, standard score; z, z score; r, raw score; h, hours.

in the current study (−1.61 and −1.57, respectively). Thus, on average, children in the current study had slightly poorer explicit phonological decoding abilities than children in *McArthur et al. (2013a)*.

## Interventions

### Specific sight word training

Children were asked to do five 30-minute sight-word training sessions per week for 8 weeks in their homes. The sight word training used an online reading program called

Literacy Planet (www.literacyplanet.com) to deliver exercises (see below) to teach children to recognise the same irregular words (see below) used by *McArthur et al. (2013a)* (Note: McArthur et al. used only two exercises to teach irregular words: one administered by computer (DingoBingo by Macroworks) and the other by a parent). Children received immediate feedback on the accuracy of their responses, which earned them points to spend on games or clothing their avatar on the LiteracyPlanet site. LiteracyPlanet also provided online access to the progress of the children in training time and performance level. This allowed the research team to detect when a child was failing to complete the required amount of training, in which case the parents were contacted to discuss how the children could better continue training.

The irregular words used in the specific sight word training delivered by Literacy Planet were selected using the following procedure: (1) REGCELEX in the CELEX database of children's written words was used to compute the rule-based pronunciations of each word in said database; (2) these pronunciations were compared to each word's dictionary pronunciation; (3) any word with a mismatch between its computer pronunciation and dictionary pronunciation was selected; (4) from this list we removed proper nouns and rude words, low frequency words that would seldom be encountered by children, and words included in CC2; (5) we ordered the words in terms of difficulty based on their written frequency, which ranged from 507073 (for *of*) down to 8 (for *scone*).

The irregular words were trained across 56 levels in Literacy Planet. Each level used eight or nine exercises to train a list of words. The lists for levels 1–30, 31–48, and 49–56 comprised 8, 14, and 24 words, respectively. The exercises for levels 1–9 included: Flash Card, Alphabetical Word Monster, Static Words, Word Snap, Floating Words, Word Finder, Word Builder 2, Spell This Word, and Word Builder 1. Levels 10–56 used the same exercises except for Word Finder. For each exercise, children were required to reach an achievement level of 80% before progressing to the next exercise. This was lower than the achievement level required by the phonics training (i.e., 100%, see below) since some of the exercises were more difficult than those in the phonics training, and we wanted the children to be able to achieve pass-rate status at a reasonable rate, and without frustration.

In the Flash Card exercise, children were asked to spell a written word that was presented on the screen and then covered. In Alphabet Monster, children were instructed to drag words presented in a list into a monster's mouth in alphabetical order. In Static Words, children were shown a static array of written words, and asked to click on a word that they heard. In Floating Words, they did the same thing except that the selection of words floated around the screen. In Word Snap, children were shown two words, and asked to click SNAP when the two words matched. In Word Finder, children were shown a matrix of letters and asked to select the first and last letter of a word that they heard. In the two Word Builder exercises, children were presented with the letters of a spoken word in mixed order, and asked to spell that word. In Spell This Word, children clicked on one of a number of bush flies on the screen, which triggered a spoken word. They were asked to spell the word.

### Specific phonics training

LiteracyPlanet was also used to deliver phonics training to the children for 30 min per day, 5 days per week, for 8 weeks in their homes. We taught phonics using nine exercises (see below) across 220 levels that increased in difficulty to train the explicit phonological decoding and encoding of consonants, short vowels, long vowels, blends, digraphs, the bossy e rule, plurals, soft 'c' and 'g,' dipthongs, 'r' sounds, and Silent Letters. No exercises included irregular words, sentences, or paragraphs of text (Note: *McArthur et al. (2013a)* used a similar number of CDROM-based computer games from Lexia Strategies for Older Students to teach children to decode and encode the same stimuli). In line with the sight word training, children received immediate feedback on the accuracy of their responses, which earned them points to spend on games or their avatar; we had online access to children's progress, allowing us to contact parents to discuss motivational strategies if children were failing to complete their training. Children were required to reach an achievement score of 100% on an exercise before moving onto the next exercise. If this was not achieved, the child repeated the exercise until they reached 100%.

In the introductory exercise—the "Movie" exercise—children were introduced to letters or letter clusters and taught their corresponding letter sound. In an "I Spy" exercise, children were presented with a number of pictures, a written letter, and a spoken letter sound. They were told that they could click on the letter to hear the letter sound, and were asked to click on the picture that started with the letter sound. In a "Letter-Sound Position" exercise, children were shown a written word and presented a spoken letter sound. They were asked to indicate whether the letter sound occurred at the beginning or end of the written word. In two "Missing Letters" exercises, children were asked to type in the missing letter of a written word. In a "Click On Words" exercise, children were shown a number of regular words, and asked to click on the word that matched a picture. In a "Bingo" exercise, children were shown a number of regular words in a matrix and were presented a spoken word. They were asked to click on the written version of the spoken word. In a "Spelling" exercise, children were asked to type in a regular word indicated by a picture. And in a "Blending" exercise, they were asked to click on letters that represent the sounds in a picture (e.g., they click on "LK" corresponding to a picture of SILK).

## Screening tests

### CC2 reading test

The CC2 comprises 40 nonwords (e.g., GRENTY), 40 irregular words (e.g., YACHT), and 40 regular words (e.g., MARSH) that increase in difficulty. The three types of stimuli were presented in an interleaved fashion on index cards. Testing for any type of item (nonwords, irregular words, regular words) was discontinued when a child made five consecutive errors for a particular type of item, or when the child reached the end of the test. A child was given 5 s to read each word before being prompted to try the next word. Scores were $z$ scores that had a mean of 0 and SD of 1.

### Nonverbal IQ

This was indexed with the Kaufman Brief Intelligence Test 2 (KBIT-2) Matrices subtest (*Kaufman & Kaufman, 2004*). Scores were standardised with a mean of 100 and an SD of 15.

### Developmental history

We used a parent questionnaire to determine if children had any known problems with their hearing, vision, neurology, or psychology, as well as establish if the children used English as their primary language at both school and home.

## Primary outcomes

### Trained and untrained irregular words

Children were asked to read aloud 58 irregular words printed on flashcards. Half of the words were included in the sight word training program ("trained irregular words") and half were not ("untrained irregular words"). Untrained irregular words were matched to the trained irregular words in terms of their written frequency, length in letters, and relative irregularity (i.e., the proportion of irregular GPCs in a word relative to the total number of GPCs in that word). Scores were total correct trained irregular words (out of 29) and total correct untrained irregular words (out of 29; Note: This is the same test used by *McArthur et al.* less two items).

### Nonword reading accuracy

This was tested using 39 untrained nonwords. A child was asked to read each nonword aloud. All items were monosyllabic, comprised 3 or 4 letters (e.g., vib, golk), and translated to two, three or four sounds. Half the items contained digraphs (e.g., th, sh), and half single-letter correspondences (e.g., t, h). Scores were total correct out of 30 (Note: McArthur et al. report that their untrained nonword test comprised 20 items but we have confirmed this was an error, and this test comprised 30 items, which were all included in the current test).

### Nonword reading fluency

We indexed nonword reading fluency using the Test of Word Reading Efficiency (TOWRE) nonword subtest (*Torgeson, Wagner & Rashotte, 1999*). This comprised 63 increasingly difficult nonwords that can be read correctly using the letter-sound rules. A child was asked to read as many nonwords as possible in 45 s. Scores were the total responses correct out of 63.

### Word reading fluency

This was tested with the TOWRE sight word subtest that comprised 104 words that increased in difficulty (*Torgeson, Wagner & Rashotte, 1999*). A child was asked to read as many words as possible in 45 s. Scores were the total responses correct out of 104.

### Reading comprehension

This was tested using the Test of Everyday Reading Comprehension (TERC) which included 10 "everyday" reading stimuli, such as a text message, a medicine label, or a shopping list (*McArthur et al., 2013b*). For each stimulus, children were asked two literal

questions based on information in the text. Scores were the total responses correct out of 20. (Note: *McArthur et al. (2013a)* used a previous version of this test that comprised an additional 3 stimuli and 6 questions.)

## Sample size

A flow diagram of the number of participants in each stage of the study is shown in Fig. 2. At the end of the study, there were 41 children in Group 1 and 44 children in Group 2.

## Sequence generation

Children were allocated to groups using minimisation randomization (balanced 1:1 for age, CC2 nonword reading, CC2 irregular word reading; executed using MINIMPY; *Saghaei, 2011*), which is considered the most appropriate sequence allocation procedure for trials comprising fewer than 100 participants. It is considered methodologically equivalent to randomization by CONSORT (*Schulz, Altman & Moher, 2010*; Note: *McArthur et al. (2013a)*) used a quasi-randomised allocation procedure).

## Allocation concealment and implementation

The lead research assistant on the project allocated children to each group and arranged their training. They concealed group allocation from research assistants who conducted the test session. All training was done online at home. All instructions to parents were provided via written documents. Parents contacted the lead research assistant if unclear about any aspect of the training.

## Blinding

Unlike drug trials, it is difficult to guarantee double blinding in cognitive treatment studies. However, parents and children were not told their group allocation, and all children received exactly the same type of training (in different orders). Most parents and children lack the expertise to discriminate between different types of reading. In addition, no tester assessed the same child twice, and no tester was aware of the child's group allocation (i.e., the tester was blind to group allocation). Thus, it is highly likely this study used a double-blind procedure.

# RESULTS

## Participant flow

A flow diagram of the number of participants in each stage of the study is shown in Fig. 2. 41 successfully completed the phonics-then-sight word training (Group 1), and 44 successfully completed the sight word-then-phonics training (Group 2). We included all children in the final analysis who completed their training, bar one child whose mother admitted at the end of the study that her child had been participating in another reading intervention. Participants who withdrew from the training did so for various personal reasons. Thus, the drop out rate in this study was low, and reasons for drop out appeared random.

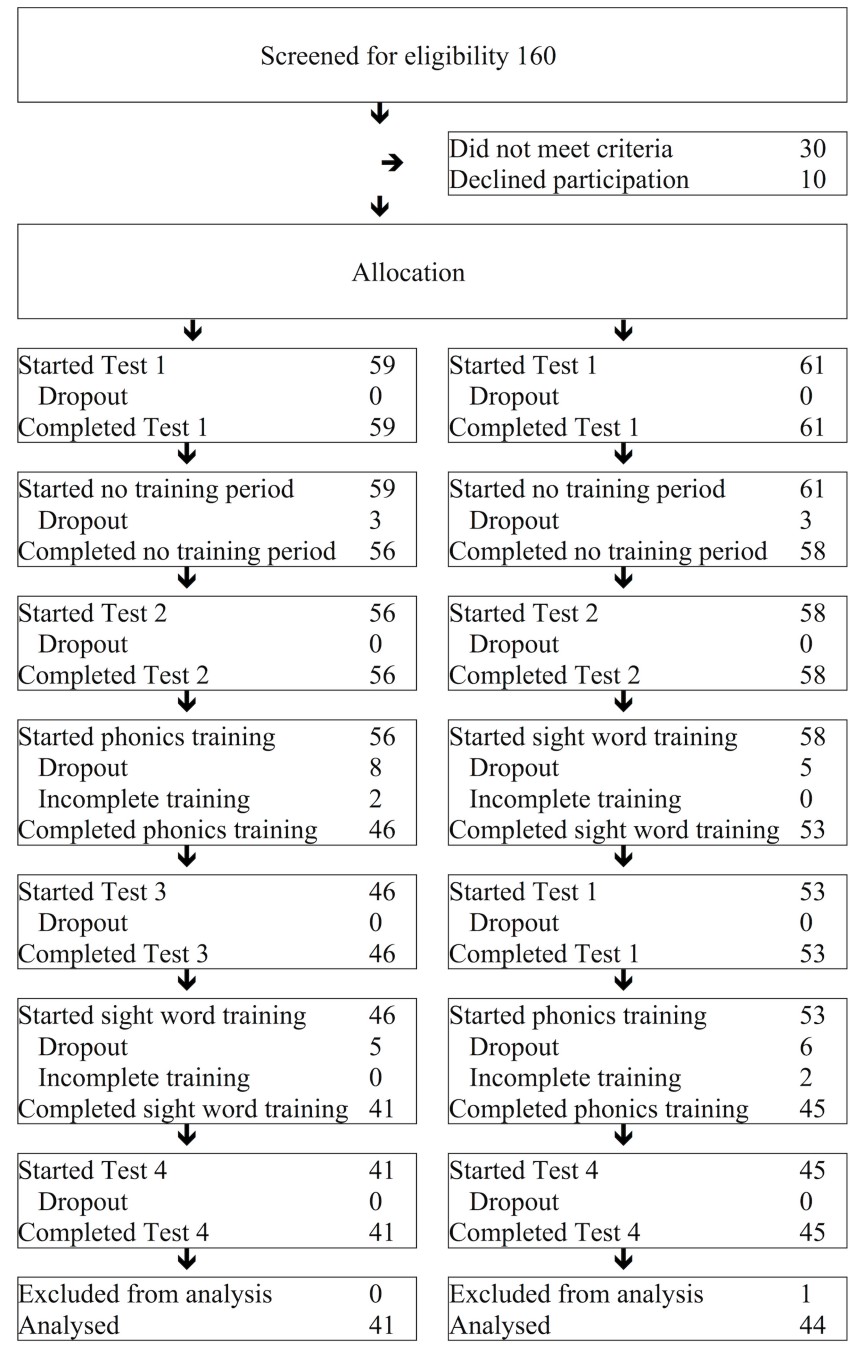

**Figure 2 Flow diagram.** The number of children who participated in each stage of the study.

## Baseline data

Between groups *t*-tests revealed that the two training groups did not differ significantly on the screening and outcome measures prior to training (i.e., see Table 2).

## Training fidelity

Based on *McArthur et al. (2013a)*, we predicted that by asking children to train for five 30-minute sessions per week for 8 weeks (20 h in total), at a minimum they would manage four 20-minute session per week for 7 weeks (due to illness, holidays, and the occasional "bad day"; a minimum of 9 h and 20 min). Two children from each group failed to reach this minimum, and were excluded from the final analysis. On average, the final sample completed around 14 h for each program (see Table 2). There was no significant difference between groups in training times.

## Numbers analysed

The analyses included 41 children in Group 1 and 44 children in Group 2. We analysed the data of participants in the groups to which they were originally allocated. In line with *McArthur et al. (2013a)*, we conducted an available case analysis on the data (i.e., based on participants with complete data).

## Outcomes

Figure 3 shows each group's mean and 95% confidence interval (CI) for gains in raw scores (i.e., difference scores) for each outcome measure. The first three CIs in each graph represent Group 1, and the last three CIs represent Group 2. Within each group, the first CI (T1T2) represents gains in raw scores from Test 1 (before training) to Test 2 (after 8 weeks of no training) due to non-training effects. The second CI (T1T3) reflects gains in raw scores between Test 1 (before training) and Test 3 (after the first 8 weeks of training). The third CI (T1T4) reflects gains in raw scores between Test 1 (before training) and Test 4 (after 16 weeks of training).

Any T1T2 CI marked with * represents a statistically significant gain due to non-training effects. Any T1T3 or T1T4 CI marked with ** represents a statistically significant gain that is significantly larger than non-training effects. Only gains marked ** were considered "valid training effects". For each effect, we calculated Cohen's *d* effect sizes calculated from the difference scores (i.e., mean group difference score/SD group difference score). Cohen's *d* scores of 0.3, 0.5, and 0.8 were considered to represent small, medium, and large effect sizes, respectively. Effect sizes for each outcome measure are compared to *McArthur et al. (2013a)* in the Table 1.

To determine if there was a reliable difference between 8 weeks of phonics and sight word training, we used a between-group ANCOVA (controlling for each group's corresponding non-training gains measured over the T1T2 no-training period) to compare T1T3 gains for Group 1 and Group 2. To determine if different orders of training had different effects on each outcome, we used a between-groups ANCOVA (controlling for non-training gains) to compare T1T4 gains for each group.

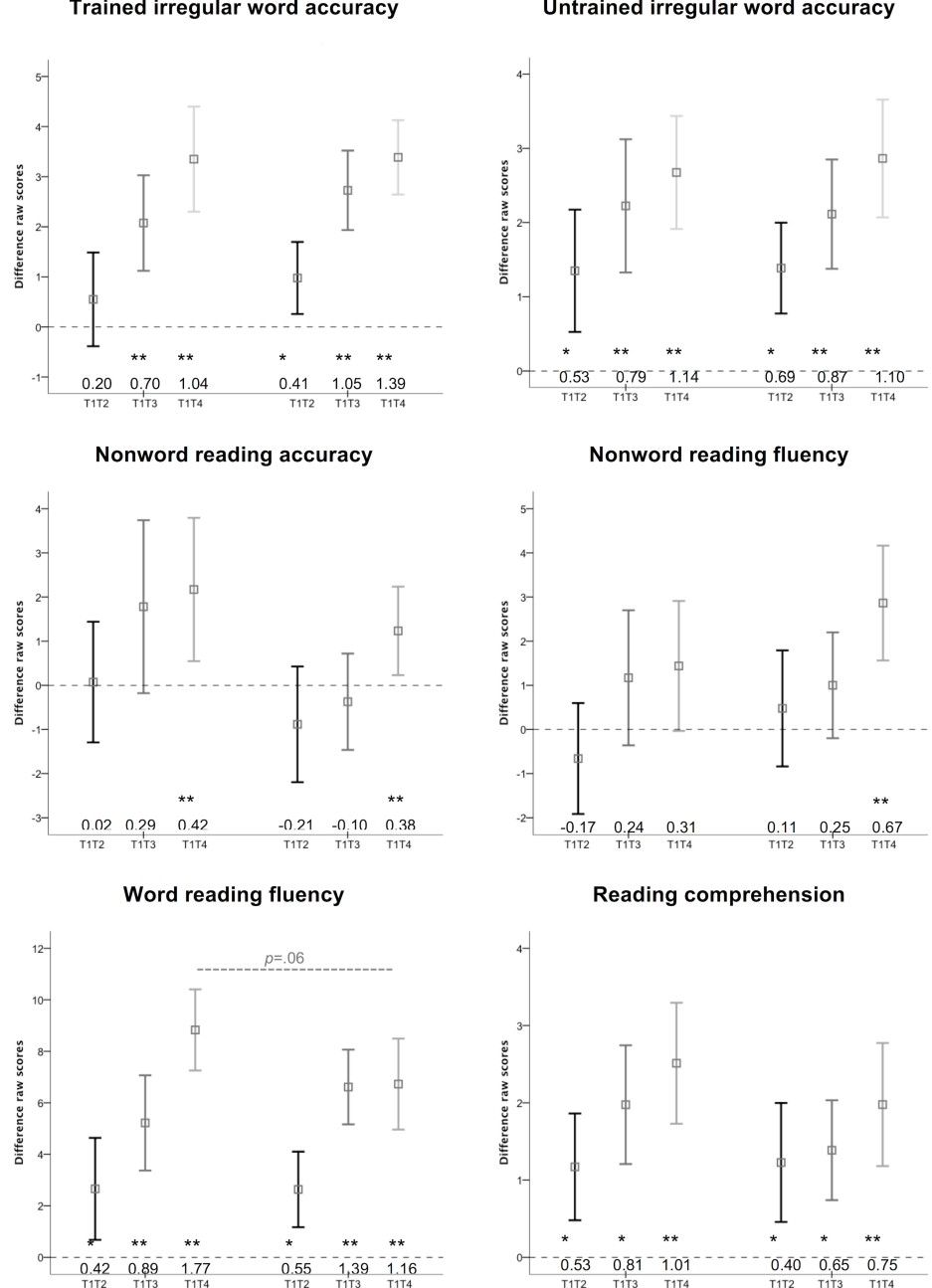

**Figure 3 Gains in outcome measures.** Group means and 95% confidence intervals for gains in raw scores for each outcome measure for the two groups.

### Trained irregular word accuracy

Eight weeks of phonics training had a moderate-to-large significant valid training effect on trained irregular words (Cohen's $d = 0.70$). Eight weeks of sight word training had a very large significant valid treatment effect on this outcome ($d = 1.05$). The difference between these effects was not statistically significant ($F(1, 81) = 0.62$, $p = .44$). Sixteen weeks of sight word and phonics training (in either order) had very large and significant

valid training effects on trained irregular word accuracy ($d = 1.04$ for Group 1 and $d = 1.39$ for Group 2). The difference between these effects was not statistically significant ($F(1, 82) = 0.21, p = .64$).

### Untrained irregular word accuracy

Eight weeks both phonics training ($d = 0.79$) and sight word training ($d = 0.87$) had large and significant valid training effects on untrained irregular word reading accuracy. The difference between these effects was not statistically significant ($F(1, 81) = 0.08, p = .78$). Sixteen weeks of phonics and sight word training (in either order) had very large and significant valid training effects on untrained irregular word accuracy (Group 1 $d = 1.14$; Group 2 $d = 1.10$). The difference between these effects was not statistically significant ($F(1, 82) = 0.12, p = .73$).

### Nonword reading accuracy

Eight weeks of phonics training had a small non-significant effect on reading untrained nonwords ($d = 0.29$). Eight weeks of sight word training had a slightly negative non-significant effect on this outcome ($d = -0.10$). The difference between these effects was not statistically significant ($F(1, 81) = 2.86, p = .10$). Sixteen weeks of phonics and sight word training (in either order) had a small-to-moderate significant valid training effect on nonword reading accuracy (Group 1 $d = 0.42$; Group 2 d $= 0.38$). The difference between these effects was not statistically significant ($F(1, 81) = 0.21, p = .65$).

### Nonword reading fluency

Eight weeks of phonics training had a small non-significant effect on nonword reading fluency ($d = 0.24$). Similarly, 8 weeks of sight word training had a small non-significant effect on the same outcome ($d = 0.25$). The difference between these effects was not statistically significant ($F(1, 82) = 1.33, p = .25$). Sixteen weeks of phonics-then-sight word training had a small non-significant effect on nonword reading fluency (0.31) while 16 weeks of sight word-then-phonics training had moderate-to-large significant valid training effect on nonword reading fluency ($d = 0.67$). The difference between these effects was not statistically significant ($F(1, 82) = 0.98, p = .32$).

### Word reading fluency

Eight weeks of phonics training had a large significant valid training effect on word reading fluency ($d = 0.89$). Eight weeks of sight word training had a very large significant valid treatment effect on this outcome ($d = 1.39$). The difference between these effects was not statistically significant ($F(1, 82) = 1.94, p = .17$). Sixteen weeks of phonics-then-sight word training had an extremely large significant valid training effect on word reading fluency ($d = 1.77$), while sight word-then-phonics training had a very large significant valid training effect ($d = 1.16$). The difference between these effects just failed to reach statistical significance ($F(1, 82) = 3.77, p = .06$).

### Reading comprehension

While 8 weeks of both phonics training and sight word training showed significant large or medium ($d = 0.81$ and $d = 0.65$, respectively) gains on reading comprehension, these

gains were not significantly larger than the non-training gains, and so were not considered valid training effects. The difference between these effects was not statistically significant ($F(1, 82) = 1.82$, $p = .18$). However, 16 weeks of phonics-then-sight word training had a very large significant valid training effect ($d = 1.01$) on reading comprehension, and 16 weeks of sight word-then-phonics training had a large significant training effect on this outcome ($d = 0.75$). The difference between these effects was not statistically significant ($F(1, 82) = 1.77$, $p = .19$).

## DISCUSSION

### Main findings

The aim of the current study was to test the replicability of the sight word and phonics training effects in poor readers reported by *McArthur et al. (2013a)*. Regarding sight word training, McArthur et al. found that specific sight word training had (1) large and significant valid treatment effects on trained irregular words (replicated in this study: $d = 1.0$), untrained irregular words (replicated in this study: $d = 0.9$), word reading fluency (replicated in this study: $d = 1.4$), and word reading comprehension (not replicated in this study: non-significant $d = 0.6$); (2) a moderate and significant valid treatment effect on nonword reading accuracy (not replicated in this study: $d = -0.1$); and (3) no valid treatment effect on nonword reading fluency (replicated in this study: non-significant $d = 0.2$). Thus, the current study replicated all bar two of the sight word training effects found by *McArthur et al. (2013a)*.

In the light of McArthur et al.'s significant and valid sight word treatment effects on reading nonwords and reading comprehension, our non-significant sight word training effects on these skills were somewhat puzzling. However, in light of dual route and triangle models of reading, these outcomes made sense. Our specific sight word training used irregular words to maximize training the lexical/semantic pathway, and minimize training the sublexical/phonological pathway. This, in turn, minimized the training of cognitive skills that underpin the ability to read nonwords (i.e., explicit phonological decoding). In addition, our sight word training, which trained the ability to read and spell irregular words by sight, did not train the types of words (typically regular words) that were used as stimuli in the reading comprehension test. Thus, the puzzle is not so much why the current study failed to find a valid sight word training effect on nonword reading and reading comprehension, but why McArthur et al. did, since they used very similar methods.

Given that *McArthur et al. (2013a)* and the current study represent the only two group controlled trials of specific sight word training (i.e., using irregular words to train the ability to recognise words from orthographic memory) in poor readers, we must turn to a single case study by *Broom & Doctor (1995)* for insight. This study examined the effect of specific sight word training (i.e., training irregular words) on reading comprehension in an 11-year-old child with developmental surface dyslexia. Like *McArthur et al. (2013a)* and the current study, specific sight word training had a significant effect on both trained and untrained irregular words. In accord with the current study, but not McArthur et al., it did not have an effect on reading comprehension. The authors suggest, and we concur,

that their specific sight word training did not generalise to reading comprehension because their training did not provide the explicit opportunity to apply newfound word reading skills in a reading-comprehension context.

Moving onto phonics training, which trained explicit phonological decoding (reading) and encoding (spelling), *McArthur et al. (2013a)* found that specific phonics training had (1) large and significant effects on trained irregular words (replicated in this study: $d = 0.7$), untrained irregular words (replicated in this study: $d = 0.8$), nonword reading fluency (not replicated in this study: $d = 0.2$), word reading fluency (replicated in this study: $d = 0.9$) and reading comprehension (replicated in this study: $d = 0.9$); and (2) a moderate-to-large effect on nonword reading accuracy (not replicated in this study: $d = 0.3$). Thus, like the sight word training, the current study replicated all bar two of the phonics training effects found by *McArthur et al. (2013a)*.

While the current study did not replicate the moderate-to-large phonics training effects on nonword reading found by *McArthur et al. (2013a)*, it is not the case that phonics had no effect on nonword reading at all. Figure 3 shows that Group 1 made gains in their nonword accuracy and fluency over their 8 weeks of phonics training that were clearly larger than their non-training gains. However, these gains just failed to reach statistical significance. After a further 8 weeks of training, Group 1's gains became statistically significant due to minor additional gains made over 8 weeks of sight word training. Group 2's data show exactly the same pattern of results but in the reverse order (i.e., because they did phonics training after sight word training). Thus, the outcomes of the current study suggest that phonics training did have an effect on nonword reading accuracy and fluency, but this effect was certainly smaller than the effect found by McArthur et al.

Why might this be the case? Since the current study used very similar methods to *McArthur et al. (2013a)*, the answer most likely lies with our sample. As noted under Participants, groups 1 and 2 in the current study had slightly weaker explicit phonological decoding abilities than groups 1 and 2 in McArthur et al. Such children may respond less well to phonics instruction (*Galuschka et al., 2014*), which would explain why the current study found smaller effects of phonics training on tests that tax phonics-related skills such as nonword reading accuracy and fluency.

Finally, in terms of order training, *McArthur et al. (2013a)* found that order of sight word and phonics training only had an effect on untrained irregular word reading, which was significantly better after phonics-then-sight word training than sight word-then-phonics training. This was not observed in the current study. The closest thing we found to an order effect was for word reading fluency, which was markedly higher after phonics-then-sight word training than the reverse. However, this order effect just failed to reach statistical significance. Combined with the outcomes of McArthur et al., this finding suggests that order of phonics and sight word training may not matter in poor readers aged from 7 to 12 years who have some phonics related skills (i.e., who can read at least a few nonwords). To a certain extent, this makes sense in terms of the dual route and triangle models of word reading, which make no predictions about the effect of training one pathway (e.g., the sublexical/phonological pathway) before another (e.g., the

lexical/semantic pathway). However, this finding does not align with the assumption that poor readers should be taught explicit phonological decoding prior to sight word reading (*Chall, 1967*). Whether or not an order effect might apply to poor readers with no phonics-related skills at all remains an empirical investigation at this point in time.

## Limitations

Because the current study is a replication of *McArthur et al. (2013a)*, it necessarily shares some of its limitations. One was the use of a "within-subjects" control group to index non-treatment gains (i.e., from Test 1 to Test 2) rather than a separate "between-subjects" untrained group (i.e., from Test 1 to Test 2 to Test 3 to Test 4). *McArthur et al. (2013a)* chose to use a within-samples control group for three reasons. First, children in a between-subjects control group may produce different (e.g., smaller) non-treatment gains than children in a within-subjects control group, which may lead to over-estimations of a treatment effect. Second, recruiting a between-subjects group would delay the administration of potentially effective treatment for poor readers for 6 months during a critical period of their reading development. And third, it is more difficult to recruit poor readers for a study in which there is a high chance of being allocated to an untreated control group.

The use of a within-subjects control group in both *McArthur et al. (2013a)* and the current study allowed the explicit measurement of non-training effects from Test 1 to 2 (T1T2), but not from Test 1 to 3 (T1T3), or from Test 1 to 4 (T1T4). Thus, the use of T1T2 gains to represent non-training gains may have underestimated true T1T3 and T1T4 non-training gains. According to previous research, non-training gains on cognitive tests over no-training periods decrease in size across test sessions (e.g., *Bartels et al., 2010*; *Collie et al., 2003*; *Kohnen, Nickels & Coltheart, 2010*). Thus, if T1T3 and T1T4 non-training gains were solely responsible for any "valid training gains" found in this study (i.e., gains marked ** in Fig. 3 that are both significantly larger than 0 and significantly larger than T1T2 gains) then (1) T1T3 gains should be less than double T1T2 gains, (2) T1T4 gains should be less then triple T1T2 gains, and (3) both groups should show very similar-sized gains (since type of training should have no effect). Examination of Fig. 3 reveals that these criteria did not apply to gains in trained irregular word accuracy, nonword reading accuracy, nonword reading fluency, or word reading fluency. This reinforces the conclusion that these gains reflect valid training gains. However, these criteria did apply to untrained irregular word accuracy and reading comprehension, which questions whether the gains in these outcomes were valid training gains, as defined by the criteria used by *McArthur et al. (2013a)*.

Given the apparently reliable effects of sight word and phonics training on trained irregular word accuracy, nonword reading accuracy, nonword reading fluency, and word reading fluency, but the questionable effects of this training on untrained irregular word accuracy and reading comprehension, it is clear that a randomised controlled trial is now needed to compare the effect of phonics and sight word training to an untrained control group and a trained control group (e.g., maths training). Since McArthur et al. and the current study both found that *order* of training phonics and sight word reading had a limited effect on outcomes in 7- to 12-year-old poor readers, such studies could focus on training

phonics and sight words in isolation. This would reduce the length of the experiment from 6 months (i.e., including a test-retest period, and two training periods) down to just 2 months (comprising a single training period). Unlike the current study, such a randomised control trial that included both an untrained control group and a trained control group would allow the explicit tracking of non-training effects across all test sessions. In the case of a trained control group, such non-training effects would include training-related Hawthorne effects, which are improvements in reading and spelling outcomes arising from an awareness of being involved in training. This new randomised controlled study may also benefit from including a passage reading test as an outcome measure to extend our understanding of the effects of phonics and sight word training in poor readers.

A second limitation of both the current study and *McArthur et al. (2013b)* was that the reading gains made by poor readers—though statistically significant, reliable, and large in effect size—did not "propel" children's reading into the average range. This does not represent a failure of phonics or sight word training. Instead, it represents the degree of difficulty of treating reading in children who are, by definition, "reading resistant." Now that we have established that phonics and sight word training both have reliable effects on heterogeneous groups of children with poor reading we can start to focus on how such effects can be maximised in children with different patterns of reading impairment.

A third limitation of this study, which was not considered by *McArthur et al. (2013a)*, and in some ways may be considered a strength, is the highly mixed educational backgrounds of the poor readers in both studies. The country in which both studies were conducted (i.e., Australia) has a highly unregulated approach to reading. The national curriculum is too vague to provide clear advice to teachers about how much time should be spent on different reading strategies, and the National Inquiry of the Teaching of Reading in Australia revealed that tertiary teaching courses allocate less than 5–10% of course time to teaching student teachers how to teach reading (*Rowe, 2005*). Adding to this confusion is the fact that over the last 5–10 years, evidence-based schools have been moving away from a strictly "whole language" approach to more mixed approaches that include phonics instruction. This means that the children in this study were receiving very different "mixes" of reading instruction at their various schools. On the one hand, this is problematic because it means that this study cannot provide any insight into how phonics training paired with sight word training might interact with different types of instruction at school. However, on the other hand, the fact that two studies (i.e., the original study and the current study) have found similar effects of phonics and sight training in groups of poor readers from very different educational backgrounds attests to the usefulness of these instructional strategies English-speaking countries where the regulation of the teaching of reading is poor.

## CONCLUSION

In sum, *McArthur et al. (2013a)* conducted the first controlled group trial to measure the effect of specific sight word training in children with poor reading, and compare the effects of specific sight word training to specific phonics training in the same. Given the

importance of discovering ways to treat poor readers to minimize their risk of academic failure and poor emotional health, combined with current concerns about the lack of direct replication of important scientific effects (*Asendorpf et al., 2013*; *Yong, 2012*), the aim of this study was the replicate the methods of McArthur et al. to test the replicability of their findings. The current study replicated the majority of McArthur et al.'s effects. Thus, the current study joins McArthur et al. in suggesting that specific sight word training paired with specific phonics training has large and significant valid treatment effects in typical samples of poor readers. It also supports the idea that poor readers should be taught to read via both phonics and sight word strategies (e.g., *Heilman, 1968*; *Nicholson, 2005*).

## ACKNOWLEDGEMENTS

We would like to thank all the children and parents for the time and effort they invested in this research. We would like to thank Shane Davis, Vicky Kadoglou, and all the people at Literacy Planet for providing the training programs for this trial, and their enthusiastic and unerring support for this project and reading research.

### Funding
This research was funded by NHMRC Project 488518 and ARC DP0879556. The funders had no role in study design, data collection and analysis, decision to publish, or preparation of the manuscript.

### Grant Disclosures
The following grant information was disclosed by the authors:
NHMRC: 488518.
ARC: DP0879556.

### Competing Interests
Associate Professor Genevieve McArthur is an Academic Editor of PeerJ.

### Author Contributions
- G McArthur conceived and designed the experiments, analyzed the data, contributed reagents/materials/analysis tools, wrote the paper, prepared figures and/or tables, reviewed drafts of the paper.
- S Kohnen conceived and designed the experiments, contributed reagents/materials/analysis tools.
- K Jones performed the experiments, contributed reagents/materials/analysis tools, reviewed drafts of the paper.
- P Eve performed the experiments.
- E Banales performed the experiments, reviewed drafts of the paper.
- L Larsen performed the experiments, contributed reagents/materials/analysis tools.

- A Castles conceived and designed the experiments, contributed reagents/materials/analysis tools, wrote the paper, reviewed drafts of the paper.

## Human Ethics

The following information was supplied relating to ethical approvals (i.e., approving body and any reference numbers):

Macquarie University Human Research Ethics Committee Ref: 5201200852.

## Data Deposition

The following information was supplied regarding the deposition of related data:

Researchers who would like access to the dataset that was analyzed for this study can submit a request to Associate Professor Genevieve McArthur via genevieve.mcarthur@mq.edu.au.

## Clinical Trial Registration

The following information was supplied regarding Clinical Trial registration:

Australian New Zealand Clinical Trials Registry (ANZCTR): 12608000454370

## Supplemental Information

Supplemental information for this article can be found online at http://dx.doi.org/10.7717/peerj.922#supplemental-information.

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
