# Peer review of "Replicability of sight word training and phonics training in poor readers: a randomised controlled trial"

_PeerJ, doi:10.7717/peerj.922_

## Round 0.1 · original submission · Major Revisions

One of the reviewers has recommended to reject the article and the other two would like you to do more work on your manuscript. I agree with the reviewers.

The main issue of concern mentioned by two of the reviewers, is the lack of a control, or comparison group to measure relative gains over time that were external to the interventions. The important question raised is whether the results can be interpreted in a meaningful way. At the very least, the conclusions are overstated, and the limitations should be acknowledged.

More information is needed in the Method section. Two reviewers note the lack of any information about previous instruction the children received in their school, and whether this instruction continued during the treatment period. Pertinent questions are raised about this and other aspects of the training in their reviews.

The manuscript needs more theoretical content which is sufficient to demonstrate how this research fits into the broader field of reading disability. Two reviewers asked for further references to support theoretical implications, including the acknowledgement of other points of view in the Introduction. One reviewer was concerned that the manuscript was a replication of a paper that was not published, and asked for further details about the similarities/dissimilarities between them. The conclusions with regard to outcome should refer to some theoretical explanation.

I would consider the article further for PeerJ, if you would address the issues raised in the reviews. I thank you for the opportunity to consider your article for publication and do hope you will find the reviews helpful for editing if you choose to do this.

With kind regards,

Claire Fletcher-Flinn, PhD
Editor, PeerJ

·

Basic reporting

The literature review was fine but in the method it would have been good to have more description of how the phonics and sight word training was done. How were the phonics rules taught? How did they train the sight words?

Experimental design

A problematic in the design is that there is no control group to take account of the effects of multiple testing and of the long time period for the study. This would have made the results more convincing. The initial no-training period of 8 weeks from T1 to T2 gives some indication of what gains might be expected but it would have been better to have a matched control group. The time 1 to time 4 gap, for example, was 16 weeks plus 8 weeks of no-training which is 24 weeks, 6 months. It might be even longer in that line 93 says the data were collected over as two year period. A control group of poor readers might make similar progress over such a long time period.
Although it is difficult to recruit control students who are poor readers, a control group could have received some other treatment such as maths which would have been a useful alternative treatment.

Validity of the findings

The results would be more convincing if there was a measure of passage reading to complement the reading comprehension measure. Word and nonword reading effects do not always transfer to passage reading.
Another question is that the raw score gains are relatively small for such a long time period. Are these meaningful gains? Were these students approaching their expected reading level after the 16 weeks? It is hard to know whether this was the case. A standardized assessment would have helped to give a reference point.
Can you say that the study actually “combined” sight word training and phonics? It was not a combination but more sequential training. A combined approach would have both kinds of training integrated together.
A possible idea for the future would be to combine phonics and sight word training with reading of text to show children how to apply their word reading skills to passage reading.

Additional comments

To summarise, the paper overstates the conclusions. It is not clear whether the results are meaningful in the absence of a proper control group. On the positive side the paper is carefully written and presented. It needs to temper its conclusions and address the limitations noted above.

·

Basic reporting

This article is a replication of an earlier study by the lead author and some co-authors. It is an experimental comparison of the effects of phonics and sight word instructional interventions on gains in various reading performances of 7-12 year-old children of low reading attainment.
In the Introduction there is a claim that it is commonly assumed poor readers should be taught phonological decoding (hence phonics) prior to sight word reading. No sources were cited. This reviewer notes that detailed accounts of the teaching of phonics have been published in several countries over the past four decades that advocate the concurrent teaching of sight word reading and phonics (e.g., Heilman, A. W., Phonics in Proper Perspective, 2nd edit. Columbus, Ohio: Charles E. Merrill, 1968. Nicholson, T. The Phonics Handbook. London, UK: Whurr, 2005).
There are several citations of references that are not list in the list of references.

Experimental design

The two types of intervention (instruction in phonics, in sight word reading) were each provided over 8 weeks by computer program in the child's home. Ample detail was provided, except for information on the types of prior reading instruction the children had experienced and what they continued to experience (e.g., at school) during the experimental intervention. The experimental instruction may complement or conflict with the child's prior and concurrent school instruction. Moreover, this information can be critical in identifying the valid range of application of the results of such an intervention study to children's instructional backgrounds.
The participants were randomly assigned to two treatment groups, one receiving 8 weeks of phonics training followed by 8 weeks of sight word training, and the other receiving these training conditions in reverse order. In both treatment groups there were initial pretests of the several reading performance variables that were followed by 8 weeks without any experimental training, and then a second pretest (comprising a repeat of the initial one). For example, in the treatment group receiving the phonics training first, in performance on accuracy of reading 29 untrained irregular words, there was a mean gain of 1.3 words reported from the first to second pretest during the untrained initial 8 weeks. There was a mean gain of 2.2 words reported from the first (not second) pretest to the first post-test 16 weeks later, following the 8 weeks of phonics training, which 16-week gain was claimed to represent the instructional gain of phonics training. There was a mean gain of 2.7 words reported from the first pretest to the second post-test 24 weeks later, comprising the 8 weeks untrained, 8 weeks of phonics, and the following 8 weeks of sight word training. This 24-week gain was claimed to represent the combined instructional effect of the phonics and the sight word training. Parallel claims were made for the other treatment group in which the order of the training was reversed.
These claims were invalid because there was no experimental control for contributions to these gains that could have occurred, over the intervals of 16 and 24 weeks, from influences external to the experimental training (e.g., concurrent or delayed effects of their school instruction). A randomly assigned control group with non-reading training may have produced reading performance gains not significantly different from those reported for either or both of the treatment groups. (The children of such a control group would not lose the opportunity for a reading instruction intervention because they would be able to receive this after the experimental period of the study.)
Under the subheading "Trial design" it was claimed that the gain from first to second pretest was an index of "non-treatment gains." Oddly, this gain was included (not excluded) from the the claimed instructional gains as above. Nonetheless, it could not control for external influences (such as school instruction) that could have occurred over 16 weeks or 24 weeks.

Validity of the findings

The experimental design problems outlined above render the claimed findings ambiguous.

Reviewer 3 ·

Basic reporting

This is an interesting paper that details a neat intervention study using two approaches common within schools. The study is actually a replication of an earlier intervention by the authors. I found the premise of the paper sound although it is unusual to see a replication for a paper that is not yet published.
There has been considerable debate in the recent literature about the meaning of “sight word reading” and at what mechanisms/potential segmentation “sight word reading” may rely upon and if “sight reading” is actually based upon a combination of phonological and orthographic knowledge rather than memorizing exceptions. The paper would appear to be taking a classic dual route approach to reading in the approach to this wider debate through use of terms such as “irregular” that have also been debated extensively in the literature. Some acknowledgement that there are other points of view about this wider literature would be useful.

Experimental design

The participants were from varied backgrounds and presumably varied schools? What was the range of teaching interventions related to reading that the children had already been exposed to? Do schools in the area sampled use phonics or sight word approaches? Might the sample have weaker phonological decoding abilities due to the teaching interventions in schools?
The paper is a replication yet uses a different teaching implementation (Literacy planet) to the original study? No data is provided on the sight words chosen for the study and why they were considered suitable or any word properties given (freq range etc) and there are no real details on the phonics training structure either. Sight word training used the “same” words as the “In Press” study yet the phonics training used “similar types” of stimuli? Could there be some more detail on these differences between the “In Press” study and this replication of how the intervention materials actually varied and if the authors have accounted for this variation in their analysis? Children in the sight word training received feedback on the accuracy of their attempts. Did the phonics based training give feedback on accuracy of learning? Did the phonics based training practice breaking down or building up of letter/clusters/words etc or just pairing as suggested? Most phonics programmes either teach the building up or breaking down of words. Did progress onto new training items in either training condition depend on being accurate in the previous training items? If so, despite the match in hours spent on the task where there any discernible differences in training items covered in the programmes and would that interact with the performance data in any predictable way?
The test items were largely the same as the replicated study but with some differences here and there. Can we be re-assured by the authors that none of these differences contributed overly to the results?

Validity of the findings

All findings seem well reported.
Might the authors speculate further on theories that best explain why there might be no order effect in terms of learning to read words? The authors speculate that there are no order effects since the sample have some phonic related skills already. However, earlier in the paper they claim that this sample have poorer decoding skills than the children in the “In press” study that did find order effects?
Table 2 is labeled Table 1 at the top of the actual table.

---

## Round 0.2 · Major Revisions

After an initial decision of rejection, we discussed this via email and I have agreed to change that rejection decision to 'major revisions'. Given that, my decision text is now as follows:

Both Reviewer 1 and 2 make the same point that you do not have an untrained control group against which to measure the treatment gains. As Reviewer 1 says you have no way of knowing whether the treatment gains are due to the treatments or to other factors. It is true that your measurements from T1 to T2 give an indication of gains from normal school instruction for these children, however, you do not know what these gains from school instruction alone might be from T2 to T3 to T4. Perhaps, school instruction alone during these time periods would have produced similar gains? You need to acknowledge this explicitly as Reviewer 1 suggests. You have not done so. In your revision, you point out an ethical reason for lack of an adequate control group, but you still need to acknowledge that you don’t know whether school instruction alone would have produced similar gains. (You also should acknowledge a ‘hawthorne effect’.)

You mention that test-retest effects tend to asymptote after a second test session. I would be surprised if this were to occur with an intervention between the two testing sessions. I can understand this occurring when there is no intervention. However, I have not read your references.

There is another point that none of the Reviewers picked up on but I think is quite important. It is the use of your terminology. Letter sounds are not ‘phonemes’, they are comprised of phonemes. For example, the letter sound for b is “bah”. There are 2 phonemes in “bah”. It is a syllable.

·

Basic reporting

You do not say exactly how the phonics rules were taught. Were there different rules taught according to level of ability, from easy to more difficult? It would be helpful to give an illustrative example of what the child did to learn a phonics rule.
You did not really explain how the sight words were trained. An illustrative example would help here of what the child had to do when you trained the sight words.
I am still not clear why the study took two years. It seems that you took on children at different periods of time so that by the time the project ended it was two years. It might help to have a table to show when children came on board over the two years.

Experimental design

You explained why you did not use a control group but there is still no way to say if the gains in the study were real or due to other factors than the training.
In the limitations it should be acknowledged that the use of a one group within subjects design is problematic in that it makes assumptions that the gains would not have occurred anyway. It can’t conclude for sure that the student gains (even though they were significant) were due to the treatments.
You have explained that the University ethics committee explicitly prohibited use of a control group. This would be your reason for a lack of a control group.
A suggestion is not to say it is unethical to have a control group. A reader could counter that with no control group the dyslexic children may have been exposed to a treatment that is not being properly tested and this would also be unethical.

Validity of the findings

You explain that the gains did not bring students up to average. Is it possible to say to what extent they improved – did they make the equivalent of 6 months gain in 6 months? I was not meaning to say that the treatment had to bring them up to average but usually you want them to improve at least to match the time they receive treatment. It would add to the validity of the findings if they improved more than the 6 months they were in the study.

Additional comments

Thank you for the considered response to my questions. As you can see I am not convinced about some of the responses and suggest that there are limitations. The results are still interesting even if you acknowledge the limitations.

·

Basic reporting

No comments.

Experimental design

The author comments for this Revision and the consequential changes to the manuscript have been carefully examined with respect to the critical issue of the PeerJ requirement of rigour and a high standard of experimental design. The design does not meet this requirement because it does not provide a way of knowing whether the reading performance gains of either of the two treatment sequences were statistically significantly greater than that obtainable from the concurrent school instruction alone.

This standard for design is not in conflict with the ethical standard required by PeerJ. Such an ethical standard is not meet by offering treatments as an addition to concurrent school instruction without obtaining rigorous evidence on whether such were more effective than the school instruction alone.

Validity of the findings

Validity is jeopardized by not meeting the PeerJ requirements for experimental design.

Reviewer 3 ·

Basic reporting

The authors have covered all the points I made in adequate detail to allow readers to make a much more measured judgement of the intervention and the context in which it took place. I am now happy to see the article published. I thank the authors for their thoughtful and detailed responses to my comments.
Vince Connelly

Experimental design

as above

Validity of the findings

as above

Additional comments

as above

---

## Round 0.3 · Major Revisions

This is a substantial improvement upon the original manuscript. However, there are a number of issues that need to be dealt with before I can accept the manuscript for publication. I outline these below:

1. The use of the terms:
a. As I've mentioned before, there are 2 forms of 'phonological recoding'. Please use the term 'explicit phonological recoding/decoding' as otherwise it is confusing to readers.
b. 'sight-word reading' is recognizing words from orthographic or lexical memory.
c. 'phoneme' - check your use of this term in the method section. Also give examples of 'speech sounds' and blending of words.
d. Hawthorne effects - misuse of this term in the section, Trial Design. You can't have such effects if there is no treatment.

2. "McArthur et al. (in press) noted that the use of within-subjects control data might be considered a conservative approach (i.e., underestimating the advantage of training over no-training) since practice effects on cognitive tests over no-training periods tend to be larger from Test 1 to Test 2 than from Test 2 to Test 3 (e.g., Bartels, Wegrzyn, Wiedl, Ackermann, & Ehrenreich, 2010; Collie, Maruff, Darby, & McStephen, 2003; Kohnen, Nickels, & Coltheart, 2010)."

Please explain how this relates to your study design and statistical comparisons, i.e., didn't you test T1T3 and T1T4?

3. "Nevertheless, the use of a within-subjects control group in both studies did not allow the explicit measurement of test-retest effects from Test 2 to 3 to 4."

The point both I, and the two Reviewers made was about factors external to the treatments, not about test-retest affects. This is a major limiting factor and must be acknowledged.

4. "It also supports the idea that all children should be taught to read via both phonics and sight word reading strategies (e.g., Heilman, 1968; Nicolson, 2005)."

This sentence is overstating your results as poor readers are not a normative sample.

5. Your treatments also included 'spelling' in addition to reading. This needs to be acknowledged in the Discussion section.

6. Please update your references as some must be published?

---

## Round 0.4 · Minor Revisions

This is a much improved manuscript, and I am satisfied that you have responded to the major issues that I raised. I thought it best to review your manuscript as a 'new' submission and read it carefully again. Here are just a few more minor corrections that I'd like you to make, as well as a few further suggestions. I outline them below:
1. Lines 37-38, 64, 65, 68, 171 quotation marks. By convention, use quotation marks for speech, i.e., "cash", "in", "pin"; use just plain letters for letters, i.e., sh, i, p; and italics for words, i.e., ship, shap, yacht, of, scone (lower case). Lines 174 -176, the quotation marks are not necessary.
2. Lines 48-50. Why mention Suggate if his work doesn't bear on your hypotheses? I would remove the reference.
3. Lines 76, 84. Please use either lexical or orthographic memory.
4. Lines 98-101; 107-109. You report that "...order had an effect on untrained irregular word reading..." then, "...this was puzzling..." Do you mean to say that the effect mentioned was the ONLY effect, so order doesn't matter? This is not clear.
5. As you note in the manuscript, this study is a replication of your previous work. I found it quite odd to see the term "reliable" in your title, and further in the manuscript. Why are you using the term "reliable" instead of "replication'? I would have thought that your previous work was reliable?
6. Line 500. Please remove "happy" as it is emotive, and thus doesn't follow APA guidelines.

---

## Round 0.5 · accepted · Accept

PeerJ production staff will now be in touch and at that point, I have a couple of edits for you, and one question where a sentence still needs clarification. Please can you deal with these directly with the production staff:

Needs correcting:

1. Lines 37-38, Remove "e.g., ship". The word ship is not a letter, or letter clusters, and in its place add "e.g., sh as in cash, etc." The word ship should be in italics and not quotation marks.
2. Line 394, Add orthographic before memory.

Query to you:
3. Line 462, Do you mean "...should have no effect"?